# Outcomes of Pediatric Liver Transplantation in Korea Using Two National Registries

**DOI:** 10.3390/jcm9113435

**Published:** 2020-10-26

**Authors:** Suk Kyun Hong, Nam-Joon Yi, Kyung Chul Yoon, Myoung Soo Kim, Jae Geun Lee, Sanghoon Lee, Koo Jeong Kang, Shin Hwang, Je Ho Ryu, Kwangpyo Hong, Eui Soo Han, Jeong-Moo Lee, Kwang-Woong Lee, Kyung-Suk Suh

**Affiliations:** 1Department of Surgery, Seoul National University College of Medicine, Seoul 08826, Korea; nobel1210@naver.com (S.K.H.); freefine@daum.net (K.C.Y.); gigsfire@gmail.com (K.H.); uishann@gmail.com (E.S.H.); lulu5050@naver.com (J.-M.L.); kwleegs@gmail.com (K.-W.L.); kssuh2000@gmail.com (K.-S.S.); 2Department of Surgery, Yonsei University College of Medicine, Seoul 03722, Korea; YSMS91@yuhs.ac (M.S.K.); DRJG1@yuhs.ac (J.G.L.); 3Department of Surgery, Samsung Medical Center, Sungkyunkwan University School of Medicine, Seoul 13557, Korea; 4hooni@gmail.com; 4Department of Surgery, Dongsan Medical Center, Keimyung University School of Medicine, Daegu 42601, Korea; kjkang@dsmc.or.kr; 5Department of Surgery, College of Medicine University of Ulsan, Asan Medical Center, Seoul 05505, Korea; shwang@amc.seoul.kr; 6Department of Surgery, Pusan National University School of Medicine, Pusan National University Yangsan Hospital, Yangsan 46241, Korea; ryujhhim@hanmail.net

**Keywords:** pediatric liver transplantation, survival, graft survival, complication

## Abstract

Background: This retrospective study aimed to evaluate overall survival and the risk factors for mortality among Korean pediatric liver transplantation (LT) patients using data from two national registries: the Korean Network Organ Sharing (KONOS) of the Korea Centers for Disease Control and Prevention and the Korean Organ Transplantation Registry (KOTRY). Methods: Prospectively collected data of 755 pediatric patients who underwent primary LT (KONOS, February 2000 to December 2015; KOTRY, May 2014 to December 2017) were retrospectively reviewed. Results: The 1-, 5-, 10-, and 15-year survival rates were 90.6%, 86.7%, 85.8%, and 85.5%, respectively, in KONOS, and the 1-month, 3-month, 1-year, and 2-year survival rates were 92.1%, 89.4%, 89.4%, and 87.2%, respectively, in KOTRY. There was no significant difference in survival between the two registries. Multivariate analysis identified that body weight ≥6 kg (*p* <0.001), biliary atresia as underlying liver disease (*p* = 0.001), and high-volume center (*p* < 0.001) were associated with better survival according to the KONOS database, while hepatic artery complication (*p* < 0.001) was associated with poorer overall survival rates according to the KOTRY database. Conclusion: Long-term pediatric patient survival after LT was satisfactory in this Korean national registry analysis. However, children with risk factors for poor outcomes should be carefully managed after LT.

## 1. Introduction

Liver transplantation (LT) is the treatment of choice for end-stage liver disease in both adults and children [1]. However, children are not just small adults and the results of pediatric LT might be somewhat different from those of adult LT. Specific disease etiologies and LT outcomes in children differ widely from those of adult patients [2]. More studies and data are needed to establish optimal independent pediatric LT guidelines.

The Korean Network Organ Sharing (KONOS) of the Korea Centers for Disease Control and Prevention was established in December 2000 to properly manage transplantation [3]. KONOS is a mandatory national data registry designed to obtain comprehensive information on waiting and allocation process before transplantation. Thus, all LT donors and recipients, regardless of LT type, are registered in KONOS. For follow-up, only patients’ deaths are recorded for annual statistics. 

The Korean Organ Transplantation Registry (KOTRY), another national organ transplantation registration system, was initiated in April 2014 [4]. The liver cohort, one of the 5 cohorts in KOTRY, consists of a central coordination unit, a medical research coordinating center, and 18 participating LT centers. Detailed data and samples (blood and tissues) from donors and recipients, including about 50% of all Korean LTs (786 of 1482 cases in 2018), are prospectively collected, registered, and followed up in KOTRY. After transplantation, follow-up occurs at 1, 6, and 12 months, and annually thereafter.

The present study aimed to determine the overall survival and risk factors for mortality among Korean pediatric patients undergoing LT using data from these two national registries.

## 2. Materials and Methods

This study was exempt from ethics approval. 

### 2.1. KONOS Database

Medical records of patients aged <18 years who underwent primary LT in Korea between 2000 and 2015 were retrospectively reviewed. The following donor data were obtained from the KONOS database: sex, age, height, body weight, blood type, and relationship to the recipient. The following recipient and donor data were obtained: sex, age, height, weight, date of LT, type of LT (living donor LT (LDLT), split deceased donor LT (DDLT), or whole liver DDLT), underlying liver disease, waiting time, ABO compatibility, relationship to recipient, hospital where the LT was performed, survival at last follow-up, and cause of death. The data on KONOS status, which is similar to the United Network for Organ Sharing (UNOS) status, were only available for DDLT, but not for LDLT. Data regarding perioperative variables and postoperative complications were not available in the database. While all LT cases are registered in the KONOS database, not many variables are collected. Eighteen centers were involved in pediatric LT in KONOS database. The period of LT was divided into three time periods: early (2000–2005), middle (2006–2010), and recent period (2011–2015) (Figure 1).

### 2.2. KOTRY Database

Medical records of patients aged <18 years who underwent primary LT between April 2014 and June 2018 were retrospectively reviewed. The following recipient and donor data were obtained from the KOTRY database: sex, age, height, body weight, date of LT, type of LT (LDLT, split DDLT, or whole liver DDLT), underlying liver disease, ABO compatibility, Child–Pugh score, pediatric end-stage liver disease (PELD) score, graft weight, pre-LT blood test, post-LT complications, survival at follow-up, and cause of death. While many pre-LT and post-LT variables are prospectively collected, only about 50% of all Korean LTs are registered in the KOTRY database. The period of LT was divided into two periods to match the KONOS database: recent (2011–2015) and most recent (2016–2018), since the KOTRY database includes LTs performed after April 2014, which is more recent compared to those included in the KONOS database (Figure 1). According to the KOTRY database, 6 centers were involved in the pediatric LT in Korea.

### 2.3. Statistical Analysis

Results are expressed as means and standard deviations or as numbers and percentages. Overall survival rates were calculated using the Kaplan–Meier method. The log-rank test and Cox proportional-hazards regression analysis were used to evaluate the association between patient characteristics and overall survival in univariate and multivariate analyses. Multivariate Cox proportional-hazards regression with backward selection was used to determine the effect of variables that were statistically significant in the univariate analysis. A *p*-value of <0.05 was considered significant. All statistical analyses were performed using SPSS version 23 (IBM Corp., Armonk, NY, USA).

## 3. Results

### 3.1. KONOS Database

There were 755 patients (333 male, 422 female; mean age 3.8 years; mean body weight 17.9 kg) aged <18 years who underwent primary LT between 2000 and 2015 (Figure 2). Pediatric LT formed 5.8% of primary LTs of Korea. The demographic and clinical characteristics of patients and donors are summarized in Table 1. More than three-quarters underwent LDLT (*n* = 584, 77.4%), while split DDLT (*n* = 86, 11.4%) and whole liver DDLT (*n* = 85, 11.3%) comprised less than 25% of cases. The number and rate of split DDLT markedly increased with time (early 0 vs. middle 8 [3.4%] vs. recent 78 [27.3%]; *p* < 0.001). Biliary atresia (46.4%) was the most common cause of LT followed by fulminant hepatic failure (9.4%) and metabolic liver disease (4.9%). ABO-incompatible LDLT was performed for 16 patients (1.9%). ABO-incompatible DDLT is not legal in Korea, even in infants. Regarding the relationship to recipient, most of the donors were mothers, followed by fathers. More than 85% of the cases were performed in high-volume centers with >10 pediatric LT cases per year.

The 1-, 5-, 10-, and 15-year survival rates were 90.6%, 86.7%, 85.8%, and 85.5%, respectively (Figure 3A). The 1-, 5-, 10-, and 15-year graft survival rates were 90.1%, 87.5%, 85.8%, and 85.3%, respectively (Figure 3B). Factors affecting patient survival are shown in Table 2. Univariate analysis showed that recipient body weight, underlying liver disease, waiting time, and center case volume were associated with overall survival rates (*p* < 0.05). Multivariate analysis identified that body weight ≥6 kg (*p* < 0.001), biliary atresia as the underlying liver disease (*p* = 0.001), and high-volume center (*p* < 0.001) were associated with better survival (Figure 4). 

### 3.2. KOTRY Database

Seventy-six patients aged <18 years (33 male, 43 female; mean age 4.9 years; mean weight 19.8 kg) who underwent primary LT between April 2014 to June 2018 were registered in the KOTRY database. The demographic and clinical characteristics of included patients and their donors are summarized in Table 1. As the KOTRY LT period was more recent compared to that of the KONOS database, 38 patients who underwent LT in the recent period, specifically between April 2014 and December 2015, were included in the KONOS database. Biliary atresia (59.2%) was the most common underlying liver disease and 47 patients (61.8%) underwent LDLT. The mean Child–Pugh score was 8.8 and the mean PELD score was 16.7. The mean duration of hospital stay was 30.9 days. The most commonly used initial postoperative immunosuppressive agent combination was tacrolimus and steroid (*n* = 68, 89.5%), followed by tacrolimus and mycophenolate mofetil (*n* = 15, 19.7%) (Appendix A). Basiliximab induction was used in 50 cases (65.8%). In Korea, mammalian target of rapamycin inhibition is not allowed for pediatric liver recipients by Korean Food and Drug Administration.

Regarding postoperative complications that required surgical, endoscopic, or radiological intervention, there were 13 (17.1%) cases of bile duct complication (bile leakage 3 and bile duct stricture 10), 7 (9.2%) cases of hepatic artery complication, 4 (5.3%) cases of intra-abdominal bleeding, 3 (3.9%) cases of portal vein complication, 3 (3.9%) cases of intra-abdominal abscess, and 2 (2.6%) cases of hepatic vein complication. The median time interval between LT and bile leakage, bile duct stricture, hepatic artery, intra-abdominal bleeding, portal vein, intra-abdominal abscess, and hepatic vein complication were 15 (range 10–18), 170 (range 10–251), 5.5 (range 1–21), 5 (range 2–15), 124 (range 55–175), 37 (range 27–109), and 9.5 (1–18) days, respectively. There were 8 (10.5%) cases of acute cellular rejection. The 1-month, 3-month, 1-year, and 2-year survival rates were 92.1%, 89.4%, 89.4%, and 87.2%, respectively. The 1-month, 3-month, 1-year, and 2-year graft survival rates were 92.0%, 87.9%, 87.9%, and 85.6%, respectively. 

There were no significant differences in overall survival and graft survival between the KONOS and KOTRY databases (Figure 3). Six patients (7.9%) died within 30 days after LT. Univariate and multivariate analysis showed that hepatic artery complication was the only factor significantly associated with overall survival rates (Table 3) (Figure 5).

## 4. Discussion

We reviewed the outcomes of Korean pediatric patients who underwent primary LT using two national databases. The KONOS database, which is a mandatory national database, included all 755 pediatric recipients who underwent primary LT between 2000 and 2015. However, it contains few variables and no information regarding postoperative complications. On the other hand, the KOTRY database consists of several variables including postoperative complications and detailed information of management outcomes. However, it includes only about 50% of all Korean LTs with the additional requirement for informed consent and permission regarding data and blood samples for prospective collection. Not all LT centers participate in the KOTRY database, limiting its representativeness as a national database. However, 6 large LDLT centers including 3 big centers performing most pediatric LTs (more than 80%) were involved in this registry. Moreover, it is limited by its relatively short follow-up since it was initiated in 2014. The present study tried to analyze national pediatric LT outcomes in Korea by combining the pros and cons of these two databases. 

The survival rates observed in the KONOS and KOTRY databases were excellent, which is consistent with the findings of previous worldwide studies on LDLT and DDLT [5,6,7,8]. Multivariate analysis showed that body weight ≥6 kg, biliary atresia as the underlying liver disease, and high-volume center were the factors independently associated with overall survival according to the KONOS database; postoperative hepatic artery complication was associated with overall survival according to the KOTRY database. Until 2016, KONOS allocated the organ according to the urgency of liver disease using the KONOS status, which was similar to the UNOS status. In 2016, KONOS adopted the PELD score for determining allocation priority. Thus, the KONOS database did not contain information regarding the PELD score in this study. In addition, because a report of the UNOS status has not been obligatory for LDLT recipients, no information of UNOS status was recorded for LDLT cases in the KOTRY database. Although the KOTRY database had both the Child–Pugh and PELD scores, these pre-transplant factors failed to show an impact on post-transplant survival. 

The KOTRY database has recent data and a relatively short follow-up. One of the limitations of our study is that these two databases had different advantages and disadvantages. However, comparing the KONOS and KOTRY databases may help evaluate the initial and recent periods of pediatric LT in Korea. Although there was no significant difference in overall survival rates between the two national databases, the clinical characteristics were different. The proportion of LDLT was lower, while that of split DDLT was higher in the KOTRY database compared to the KONOS database. In proportion, there were more children with metabolic liver disease as underlying disease, which was the second most common indication of pediatric LT, followed by biliary atresia, and ABO-incompatible cases in the KOTRY database compared to the KONOS database (*p* < 0.05). These differences align well with the recent trend in Korean LT including adult cases (Figure 2) [9,10,11,12].

It has been reported that young age and/or low body weight are associated with high rates of morbidity and mortality [13,14,15,16,17,18,19,20,21]. Several studies reported an increased risk of infection or vascular complications in young and/or low-body-weight children [13,14,15,16]. The poorer result in these young and/or low-body-weight children can also be explained by the inevitable use of relatively large-for-size grafts, which results in insufficient tissue oxygenation and graft compression [13]. Considering that liver disease in children is associated with malnutrition and growth retardation, low body weight, rather than young age, may better reflect the size of the patient and be more accurately related to the poorer outcome [13]. The present study showed similar results. A body weight <6 kg was associated with poorer overall survival in the KONOS database, while age showed no significant association with overall survival; however, the KOTRY database, which includes more recent data, showed no such findings. The cutoff value of body weight was based on previous publications [11,16].

Patients with biliary atresia had a better 15-year survival rate of 90.2% compared to patients with other underlying liver diseases. Similar findings were observed in previous reports including LDLT and DDLT series [6,8]. According to the KONOS database, patients with biliary atresia were younger and lighter. Although not statistically significant, patients with biliary atresia had tendency of having longer waiting time compared with patients with other underlying liver disease. The better survival in patients with biliary atresia can be also explained in conjunction with the result from the KOTRY database which showed that patients who experienced postoperative hepatic artery complication had poorer survival, especially in Korea, where more than 75% of pediatric LT cases are LDLT. According to the KOTRY database, the proportion of split DDLT has increased. However, in most split DDLTs in Korea, the left lateral section graft takes the left hepatic artery and portal vein similar to LDLT regarding hilar division [11]. In addition, the proportion of metabolic liver disease as the underlying disease serving as the indication for pediatric LT increased over time in the KOTRY database. We previously reported that patients with biliary atresia had a lower rate of postoperative hepatic artery complications after LT compared to metabolic liver disease, owing to the relatively larger diameter of the proper hepatic artery with portal hypertension, which made anastomosis easier, in addition to the relatively good quality of the dilated hepatic artery [11]. For this reason, the rate of hepatic artery complication is considered to be higher than noted in previous reports [14,22].

Univariate analysis showed that patients who had longer waiting time (≥30 days) had better survival (Appendix A). Patients who underwent LT because of fulminant hepatic failure had a mean waiting time of 6.0 days and the most common transplantation type for fulminant hepatic failure was LDLT (90.1%). However, more patients had waiting time >30 days in biliary atresia compared with other underlying diseases. Patients with life-threatening liver failure inevitably underwent LT quickly to shorten the waiting time and to improve survival rate. Thus, patients with generally better clinical condition had longer waiting times. The type of LT (LDLT vs. split DDLT vs. whole liver DDLT) did not affect mortality. However, when comparing LDLT with DDLT (split DDLT or whole liver DDLT together), LDLT showed poorer survival compared with DDLT (Appendix A). Patients who cannot afford to wait usually undergo LDLT in Korea due to the shortage of deceased organ, especially for children. Patients who underwent LDLT had shorter mean waiting time than patients who underwent DDLT (74.3 days vs. 274.1 days; *p* < 0.001). LDLT using the reduced left lateral section graft plays an important role as a good alternative for children with critical conditions at the time of LT. With critically ill patients undergoing emergency LDLT, patients with better clinical condition could fortuitously be able to undergo DDLT. When performing split procurement, the in situ split technique is usually employed resulting in a short ischemic time. Cold ischemia is a well-known donor-related risk factor that affects graft and patient survival [23,24,25,26]. Generally, DDLT has a longer cold ischemic time than LDLT. However, with the in situ split procedure and short transport distance in Korea (driving end to end in Korea takes <6 h), DDLT might deliver fewer adverse outcomes than expected. 

Some reports have focused on better outcomes in children receiving maternal grafts compared to those receiving other grafts [27,28]. Tolerance to non-inherited maternal antigens can be promoted by the presence of maternal cells in offspring, resulting in lower graft failure [27]. Data regarding the relationship to the recipient were available in both the KONOS and KOTRY databases. Both databases failed to show better survival in children that received livers from their mother compared to those that received livers from others. However, focusing on 584 LDLT cases in the KONOS database, 241 children who received the liver from their mother showed significantly better survival compared to 343 children who received the liver from others (*p* = 0.031).

There has been notable improvement in surgical techniques and medical treatment over the past two decades. A Korean multicenter study that analyzed 502 children who underwent LT between 1988 and 2010 showed superior overall and graft survival rates in patients after 2003 compared to those before 2003 [29]. The present study, which has a larger sample size and includes more recent cases since 2000, failed to show an overall survival improvement according to the period of LT. However, there was significantly less mortality within 1 month in the recent period compared to the early (*p* = 0.023) and middle (*p* = 0.015) periods. Expansion of disease criteria and graft type, including an ABO-incompatible graft from a living donor and a split graft from a deceased donor, may have affected the result. There was no split DDLT in the early era, while there were 8 cases in the middle era, and 78 cases more recently. Among 16 cases of ABO-incompatible LT, 14 were performed in the recent era. 

One of the strongest risk factors for mortality after pediatric LT was low-volume center. Three centers performed more than 10 cases yearly and conducted more than 85% of all pediatric LTs in Korea. A similar result was identified by previous study which used the database of Korean National Healthcare Insurance Service [30]. According to that study, centers with more than 10 cases of pediatric LT yearly had better short-term and long-term survival compared to centers with less than 10 cases annually [30]. Centralization or early referral to specialized centers has been suggested for superior patient outcomes, especially for pediatric hepatobiliary disease [8,30,31,32]. Pediatric LT centralization to centers with specialized programs and teams may be needed for better survival, especially in children with risk factors related to poor survival (i.e., body weight <6 kg or other underlying diseases other than biliary atresia). 

This study had several limitations and some of them have already been mentioned. First, both databases showed unique weaknesses; for example, fewer variables were recorded in the KONOS database than in the KOTRY database. In contrast, the KOTRY database was only recently implemented and, therefore, had limited patient data compared to the KONOS database. Second, we observed duplication of data as 38 patients were listed in both databases, which was a structural weakness that may have influenced the accuracy of our findings when we compared the two databases. However, we believe our findings are still valid because they may highlight an era effect. Third, as this was a retrospective study, some relevant data were missing for analysis (e.g., data on pretransplant waiting time). Last, the information in the KONOS and KOTRY databases were acquired from each transplant center. However, this study was the first in Korea to include such a large sample size and to evaluate all Korean pediatric LT outcomes using two main national registry databases. We believe that there would be benefit in merging the two databases into one efficient database that would enable analysis of more pre-, intra-, and post-LT factors affecting outcomes to further refine pediatric LT procedures. Therefore, we are currently examining how these databases can be unified to help better analysis of variables in future studies. 

## 5. Conclusions

Korean pediatric LT showed excellent overall survival, which is consistent with those of previous worldwide reports. However, children with body weight <6 kg and without biliary atresia as the underlying liver disease had a higher risk of poor overall survival. Physicians should pay more attention to these patients. In addition, centers with lower case volumes (≤10 pediatric LTs per year) had poorer outcomes after pediatric LT compared to centers with higher case volumes. High-risk patients should be carefully reviewed with respect to early referral to a specialized center. Postoperative hepatic artery complication was still related to poor survival. Thus, risk factor analysis of hepatic artery complications in a large cohort and more meticulous surgery are required in pediatric LT.

## Figures and Tables

**Figure 1 jcm-09-03435-f001:**
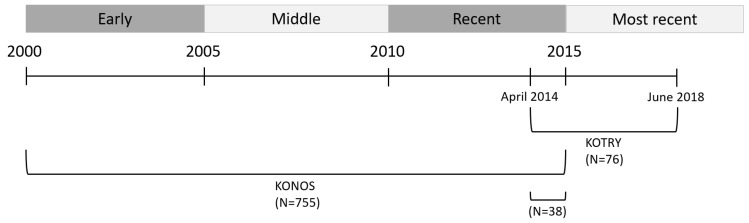
Schematic graph showing chronological difference between the Korean Network Organ Sharing (KONOS) and Korean Organ Transplantation Registry (KOTRY) databases.

**Figure 2 jcm-09-03435-f002:**
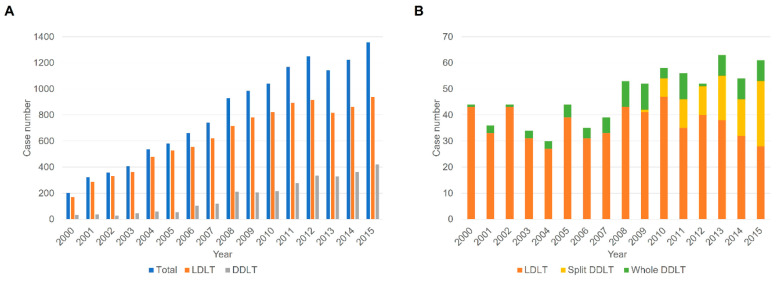
Annual number of (**A**) total liver transplantations (LT) including adult and children and (**B**) pediatric LT alone.

**Figure 3 jcm-09-03435-f003:**
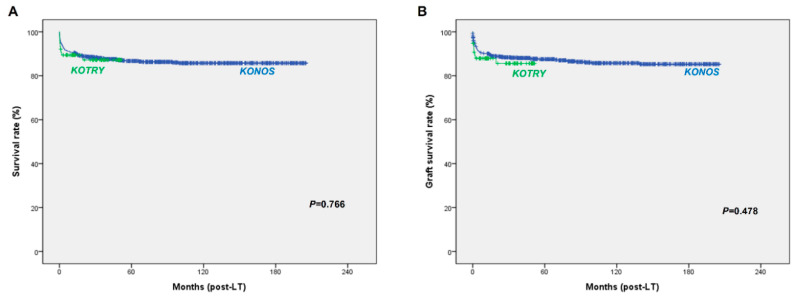
Kaplan–Meier analysis of survival using the KONOS and KOTRY databases. (**A**) Patient survival, (**B**) graft survival.

**Figure 4 jcm-09-03435-f004:**
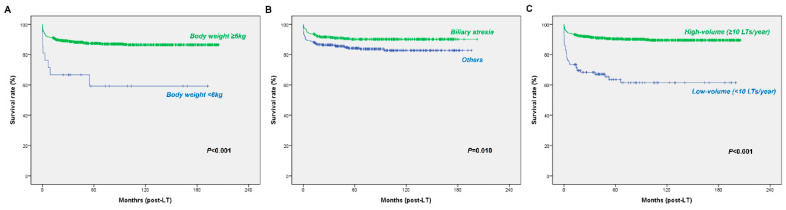
Kaplan–Meier analysis of overall survival using the KONOS database according to (**A**) body weight, (**B**) underlying disease, and (**C**) center case volume.

**Figure 5 jcm-09-03435-f005:**
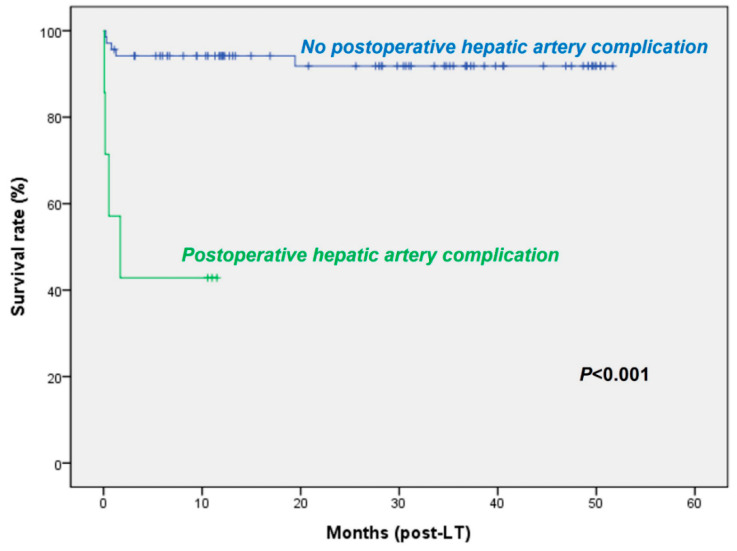
Kaplan–Meier analysis of overall survival using the KOTRY database according to postoperative hepatic artery complication.

**Table 1 jcm-09-03435-t001:** Demographic and clinical characteristics of pediatric patients in the KONOS and KOTRY databases.

Variables	KONOS *n* = 755	KOTRY *n* = 76	*p*-Value
Sex, male:female	333:422	33:43	1.000
Age, mean ± SD, years	3.8 ± 5.1	4.9 ± 5.5	0.089
Height, mean ± SD, cm	97.1 ± 36.2	97.4 ± 36.3	0.949
Body weight, mean ± SD, kg	17.9 ± 16.4	19.8 ± 18.1	0.323
Period of LT, *n* (%)			<0.001
Early (2000–2005)	232 (30.7)	-	
Middle (2006–2010)	237 (31.4)	-	
Recent (2011–2015)	286 (37.9)	38 (50.0)	
Most recent (2016–2018)	-	38 (50.0)	
Type of LT, *n* (%)			
LDLT	584 (77.4)	47 (61.8)	0.003
Split DDLT	86 (11.4)	23 (30.3)	<0.001
Whole liver DDLT	85 (11.3)	6 (7.9)	0.371
Underlying liver disease, *n* (%)			
Biliary atresia	350 (46.4)	45 (59.2)	0.032
Fulminant	71 (9.4)	9 (11.8)	0.492
Metabolic disease	37 (4.9)	9 (11.8)	0.029
Malignancy	23 (3.0)	1 (1.3)	0.716
PBC or PSC	8 (1.1)	3 (3.9)	0.071
HBV or HCV	8 (1.1)	0	1.000
Others	151 (20.0)	5 (6.6)	0.004
Unknown	12 (1.6)	3 (3.9)	0.151
Missing	95 (12.6)	0	0.002
Waiting time, mean ± SD, days	119.6 ± 325.6	NA	NA
ABO-incompatible, *n* (%)	16 (2.1)	8 (10.5)	0.001
Preoperative blood test			
Albumin, mean ± SD, g/dL	NA	3.4 ± 0.5	NA
Bilirubin, mean ± SD, mg/dL	NA	11.0 ± 10.0	NA
PT INR, mean ± SD	NA	1.9 ± 1.2	NA
Creatinine, mean ± SD, mg/dL	NA	0.4 ± 0.9	NA
Na, mean ± SD, mmol/L	NA	139.1 ± 4.3	NA
Child–Pugh score, mean ± SD	NA	8.8 ± 2.1	NA
PELD score, mean ± SD	NA	16.7 ± 11.9	NA
Center case volume, *n* (%)			
High-volume, ≥10 LTs/year	646 (85.6)	NA	NA
Low-volume, <10 LTs/year	109 (14.4)	NA	NA
Donor sex, male:female	37: 80	32:4	0.209
Donor age, mean ± SD, years	31.1 ± 10.2	32.8 ± 11.5	0.186
Donor height, mean ± SD, cm	160.5 ± 21.4	162.2 ± 14.8	0.511
Donor body weight, mean ± SD, kg	59.8 ± 15.7	61.0 ± 14.8	0.538
Graft weight, mean ± SD, g	NA	335.6 ± 177.7	NA
GRWR, mean ± SD, %	NA	2.3 ± 1.1	NA
Relationship to recipient			
Mother	241 (31.9)	31 (40.8)	0.116
Father	186 (24.6)	12 (15.8)	0.084
Sibling	11 (1.5)	1 (1.3)	1.000
Other relatives	54 (7.1)	2 (2.6)	0.134
Non-relatives	181 (24.0)	30 (39.5)	0.003
Missing	82 (10.9)	0	1.000
Preoperative blood test			
Anti-HBc (+), *n* (%)	NA	13 (17.1)	NA
AST	NA	61.1 ± 51.1	NA
ALT	NA	49.1 ± 39.1	NA
Bilirubin	NA	1.1 ± 0.8	NA
Hospital stay, mean ± SD, days	NA	30.9 ± 19.2	NA
Surgical complication, *n* (%) ^a^	NA	21 (27.6)	NA
Intra-abdominal bleeding ^b^	NA	4 (5.3)	NA
Hepatic artery ^b^	NA	7 (9.2)	NA
Portal vein ^b^	NA	3 (3.9)	NA
Hepatic vein ^b^	NA	2 (2.6)	NA
Biliary ^b^	NA	13 (17.1)	NA
Intra-abdominal abscess ^b^	NA	3 (3.9)	NA
Acute cellular rejection, *n* (%)	NA	8 (10.5)	NA

SD, standard deviation; NA, not available; LT, liver transplantation; LDLT, living donor liver transplantation; DDLT, deceased donor liver transplantation; PBC, primary biliary cirrhosis; PSC, primary sclerosing cholangitis; HBV, hepatitis B virus; HCV, hepatitis C virus; PELD, pediatric end-stage liver disease; GRWR, graft-to-recipient weight ratio; Anti-HBc, hepatitis B core antibody; AST, aspartate aminotransferase; ALT, alanine aminotransferase; ^a^ Number of patients who had complications; ^b^ Cases of complications.

**Table 2 jcm-09-03435-t002:** Univariate analysis of factors associated with patient survival using the KONOS database.

Variables			*n*	1 Year Survival Rate (%)	5 Year Survival Rate (%)	10 Year Survival Rate (%)	*p*-Value
Sex							0.740
	Male		333	89.8	87.2	86.7	
	Female		422	91.2	86.3	84.9	
Age, year							0.564
	<1		271	89.3	85.2	85.2	
	≥1		484	91.3	87.6	86.0	
Body weight, kg							<0.001
	<6		21	66.7	59.3	59.3	
	≥6		732	91.3	87.5	86.5	
Period of LT							0.276
	Early (2000–2005)		232	90.1	85.8	84.1	
	Middle (2006–2010)		237	87.8	84.8	84.8	
	Recent (2011–2015)		286	93.4	89.6	89.6	
Type of LT							0.131
	LDLT		584	89.4	85.5	84.4	
	Split DDLT		86	96.5	91.7	91.7	
	Whole liver DDLT		85	92.9	90.0	90.0	
Underlying disease							
	BA vs. others						0.010
		BA	350	93.7	90.2	90.2	
		Others	310	88.4	84.3	82.8	
	Fulminant vs. others						0.340
		Fulminant	71	85.9	84.5	84.5	
		Others	589	91.9	87.8	87.1	
	Metabolic disease vs. others						0.206
		Metabolic disease	37	97.3	94.6	94.6	
		Others	623	90.9	87.0	86.4	
	Malignancy vs. others						0.479
		Malignancy	23	91.3	80.7	80.7	
		Others	637	91.2	87.6	87.0	
Waiting time, days							0.009
	<30		408	88.7	84.5	83.6	
	≥30		323	93.5	91.3	90.1	
ABO compatibility							0.422
	Compatible		739	90.8	86.9	85.9	
	Incompatible		16	81.3	81.3	81.3	
Relationship to recipient							0.168
	Mother		241	92.1	89.4	88.4	
	Other than mother		514	89.9	85.5	84.5	
Center case volume							<0.001
	High-volume (≥10 LTs/year)		646	93.5	90.4	89.6	
	Low-volume (<10 LTs/year)		109	73.4	63.6	61.5	
Donor sex							0.770
	Male		375	91.5	86.9	86.0	
	Female		380	89.7	86.6	85.6	
Donor age, years							0.438
	<18		70	91.4	89.7	89.7	
	≥18		685	90.5	86.4	85.4	
Donor body weight, kg							0.305
	<40		49	93.9	91.5	91.5	
	≥40		706	90.4	86.4	85.4	

**Table 3 jcm-09-03435-t003:** Univariate analysis of factors associated with patient survival using the KOTRY database.

Variables	*n*	6 Month Survival Rate (%)	1 Year Survival Rate (%)	2 Year Survival Rate (%)	*p*-Value
Sex					0.158
	Male	33	97	97	91.6	
	Female	43	83.7	83.7	83.7	
Age, year					0.537
	<1	27	85.2	85.2	85.2	
	≥1	49	91.8	91.8	88.3	
Body weight, kg					0.344
	<6	4	75	75	75	
	≥6	72	90.2	90.2	87.9	
Period of LT					0.372
	Recent (2011–2015)	38	84.2	84.2	84.2	
	Most recent (2016–2018)	38	94.7	94.7	82.9	
Type of LT					0.172
	LDLT	47	93.6	93.6	89.7	
	Split DDLT	23	78.3	78.3	78.3	
	Whole liver DDLT	6	100	100	100	
Underlying disease					
	BA vs. others					0.097
		BA	45	93.2	93.2	93.2	
		Others	31	83.9	83.9	78.9	
ABO compatibility					0.299
	Compatible	68	88.2	88.2	85.8	
	Incompatible	8	100	100	100	
Child–Pugh score					0.766
	A	11	90.9	90.9	90.9	
	B, C	65	89.2	89.2	86.6	
PELD score					0.33
	<20	38	89.5	89.5	89.5	
	≥20	26	84.6	84.6	78.1	
Relationship to recipient					0.576
	Mother	31	93.5	93.5	88.4	
	Other than mother	45	86.6	86.6	86.6	
Donor sex					0.325
	Male	32	87.5	84.3	84.3	
	Female	44	93.2	93.2	89.6	
Donor age, years					0.369
	<18	6	100	100	100	
	≥18	70	88.5	88.5	86.1	
Donor body weight, kg					0.411
	<40	5	100	100	100	
	≥40	71	88.7	88.7	86.3	
GRWR, %					0.797
	<2	19	94.7	94.7	94.7	
	≥2 and <4	21	95.2	95.2	88.4	
	≥4	5	100	100	100	
Hepatic artery complication					<0.001
	No	69	94.2	94.2	91.8	
	Yes	7	42.9	42.9	42.9	
Portal vein complication					0.542
	No	73	89	89	86.7	
	Yes	3	100	100	100	
Hepatic vein complication					0.633
	No	74	89.1	89.1	86.9	
	Yes	2	100	100	100	
Biliary complication					0.666
	No	63	88.9	88.9	86.5	
	Yes	13	92.3	92.3	92.3

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
