# Peer review of "Outcomes of Pediatric Liver Transplantation in Korea Using Two National Registries"

_jcm, 2020, doi:10.3390/jcm9113435_

Round 1

Reviewer 1 Report

Summary

This article examines the survival rates and risk factors associated with pediatric liver transplant. Data used from two different national registries from Korea. The majority of included patients came from the KONOS registry which had few variables and post-op data. A small number of patients came from the recent KOTRY registry. Reported survival rates were similar. Risk factors were identified based data from either of the registries (no combined data was used). The main factors associated with increased survival based on KONOS registry was body weight >6kg, biliary atresia as underlying liver disease, and transplant occurring at high-volume center. KOTRY registry identified hepatic artery complications post-transplant with poorer survival rate.

Broad comments:

  • Well written paper with large sample size that shows good overall survival rates for post-liver transplant patients in this country
  • Results are presented from either KONOS registry or KOTRY registry but no combined data was used. Is there a reason for this? Although KONOS database had fewer available variables, there were many overlaps (including body weight which was not significant in KOTRY database)
  • Limited variables in KONOS (majority of patients included in the study) is a weakness, which is stated by the authors
  • Body weight cutoff used was 6kg. No explanation was given as to why this weight was used
  • No time frame of post-transplant complications were given. Was this information available in the KOTRY database? Time frame of post-op complications would be important to know when looking at risk factors for later survival
  • In discussion, authors state that LDLT had poorer survival compared to total DDLT. One explanation given is shorter waiting times for LDLT but no data is given to support that. Does the data from KONOS show that wait time is shorter for LDLT compared to DDLT? If so, should show that.
  • In the conclusion, authors state that patients <6kg and without biliary atresia as underlying liver disease had higher risk of poor overall survival and therefore should receive “more intensive care”. Please expand on what that means as all post-liver transplant patients typically receive close monitoring and care

Specific comments:

  • Discussion line 255-257. Discrepancy between last two sentences of that paragraph: Both 254 databases failed to show better survival in children that received livers from their mother compared 255 to those that received livers from others. However, focusing on LDLT in the KONOS database, 256 children who received the liver from their mother showed significantly better survival compared to 257 those who received the liver from others. 258 There has been notable improvement.”
    • If last sentence is true, please provide data to support that.
    • Maternal donor vs other donors for LDLT is more relevant that maternal donor vs all others (including DDLT).

Author Response

Broad comments:

Well written paper with large sample size that shows good overall survival rates for post-liver transplant patients in this country

Results are presented from either KONOS registry or KOTRY registry but no combined data was used. Is there a reason for this? Although KONOS database had fewer available variables, there were many overlaps (including body weight which was not significant in KOTRY database)

--> While many pre-LT and post-LT variables are prospectively collected, only about 50% of all Korean LTs are registered in the KOTRY database. According to the KOTRY database, only 6 centers were involved in the pediatric LT in Korea. Thus, it is less representative. With this less representativeness, simply combining the data would result in bias.

Limited variables in KONOS (majority of patients included in the study) is a weakness, which is stated by the authors

Body weight cutoff used was 6kg. No explanation was given as to why this weight was used

--> The cutoff value of body weight was based on previous publications including ours. We have added the sentence in the “Discussion” section as below.

“The cutoff value of body weight was based on previous publications [11,16].”

No time frame of post-transplant complications were given. Was this information available in the KOTRY database? Time frame of post-op complications would be important to know when looking at risk factors for later survival

-->We have added the following sentence in the “Results” section as below;

“The median time interval between LT and bile leakage, bile duct stricture, hepatic artery, intra-abdominal bleeding, portal vein, intra-abdominal abscess, and hepatic vein complication were 15 (range 10-18), 170 (range 10-251), 5.5 (range 1-21), 5 (range 2-15), 124 (range 55-175), 37 (range 27-109), and 9.5 (1-18) days, respectively.”

In discussion, authors state that LDLT had poorer survival compared to total DDLT. One explanation given is shorter waiting times for LDLT but no data is given to support that. Does the data from KONOS show that wait time is shorter for LDLT compared to DDLT? If so, should show that.

-->Thank you for the valuable comment. We additionally analyzed the waiting time in LDLT and DDLT. As you expected, the waiting time was shorter in LDLT patients compared to DDLT patients. We have added the sentence in the “Discussion” section as below;

“Patients who underwent LDLT had shorter mean waiting time than patients who underwent DDLT (74.3 days vs. 274.1 days; P<0.001).”

In the conclusion, authors state that patients <6kg and without biliary atresia as underlying liver disease had higher risk of poor overall survival and therefore should receive “more intensive care”. Please expand on what that means as all post-liver transplant patients typically receive close monitoring and care

-->We meant that more attention is needed. We changed the sentence to clear our meaning as below;

“Physicians should pay more attention to these patients.”

Specific comments:

Discussion line 255-257. Discrepancy between last two sentences of that paragraph: Both 254 databases failed to show better survival in children that received livers from their mother compared 255 to those that received livers from others. However, focusing on LDLT in the KONOS database, 256 children who received the liver from their mother showed significantly better survival compared to 257 those who received the liver from others. 258 There has been notable improvement.” If last sentence is true, please provide data to support that.

Maternal donor vs other donors for LDLT is more relevant that maternal donor vs all others (including DDLT).

-->According to KONOS database, 584 children underwent LDLT. When focusin on these LDLT patients only, 241 children who received the liver from their mother showed better survival compared to 343 children who received the liver from others. We changed the sentence in the “Discussion” section as below;

“However, focusing on 584 LDLT cases in the KONOS database, 241 children who received the liver from their mother showed significantly better survival compared to 343 children who received the liver from others (P=0.031).”

Reviewer 2 Report

Hong et al. present the results of pediatric liver transplantation in Korea.   The greatest novelty of this study is the presentation of outcomes for liver transplantation in Korea, thereby describing the longterm post transplant outcomes, the complications, and the trends regarding organ donor type (ie. LDLT vs. DDLT).  This paper is significantly limited as a descriptive paper without any more deeper analysis, so that their conclusions are significantly limited and contribute very little to known literature. I have the following major concerns:

  1. why do the authors choose to make the fact that there are 2 different registries (with a limited overlapping time period) such a central part of this paper?   Why did the authors chose not to just merge the 2 registries, and report this as: Pediatric liver transplantation in South Korea -- retrospective analysis from the transplant registries in South Korea.  For that matter, how were the overlapping 38 patients treated?   Were they counted double in the 2 different registries?  or were the data merged?  The choice of comparing patients in 2 different datasets makes no sense.   Could there be a era effect? 

2. Could the authors provide waiting list analysis?  How many patients were listed, what was waitlist mortality, and was there any trend towards improved waitlist outcomes incorporating more deceased donor liver transplants and splitting livers?   

3. the authors suggest that LDLT had worse outcomes due to higher "acuity" patients.   Could the authors have performed a subgroup propensity analysis to determine that this is really true?   If we controlled for equal PELD, equal acuity patients on the waiting list, would outcomes for LDLT vs. DDLT be the same?

4. the design of this study does not support their conclusions.   While the observation of improved outcomes for higher volume centers is made, this is not appropriately studied or presented by the authors.   A more adjusted analysis should have been done comparing outcomes for high volume centers vs. low volume centers, and their data presented.   This analysis should have also been done for smaller bodyweight recipients and patients without biliary atresia.   It is not enough to do the univariate analysis to arrive at this conclusion. 

Author Response

Hong et al. present the results of pediatric liver transplantation in Korea. The greatest novelty of this study is the presentation of outcomes for liver transplantation in Korea, thereby describing the longterm post transplant outcomes, the complications, and the trends regarding organ donor type (ie. LDLT vs. DDLT). This paper is significantly limited as a descriptive paper without any more deeper analysis, so that their conclusions are significantly limited and contribute very little to known literature. I have the following major concerns:

1.why do the authors choose to make the fact that there are 2 different registries (with a limited overlapping time period) such a central part of this paper? Why did the authors chose not to just merge the 2 registries, and report this as: Pediatric liver transplantation in South Korea -- retrospective analysis from the transplant registries in South Korea. For that matter, how were the overlapping 38 patients treated? Were they counted double in the 2 different registries? or were the data merged? The choice of comparing patients in 2 different datasets makes no sense.   Could there be a era effect?

-->KONOS database has many missing data and no follow-up detailed data except patients and organ survival although this is the only mandatory national registry of every organ transplantation. Thus, we used the data from KOTRY for the most recent period; KOTRY data was prospectively collected and detailed follow-up data can be used. It was not possible to simply merge the two registry data because the variables included in the two registries were different.

In fact, we didn’t plan to compare the survival of the two databases initially. However, while summarizing the demographics of the two databases, we thought it would be additionally interesting to compare them due to following reasons.

In terms of the whole, representing all the pediatric LT cases in Korea, the KONOS database fits well. However, it has limitation of lack of data regarding perioperative variables and postoperative complications. Moreover, the update of the data together with the survival outcome is slow and inevitably used data including pediatric LT cases performed between 2000 and 2015.

On the other hand, the KOTRY database includes many pre-LT and post-LT variables. The KOTRY registration system was initiated in 2014. Thus, the KOTRY database includes more recent cases compared to the KONOS database. However, only about 50% of all Korean LTs were collected, registered, and followed up.

Although there are some limitations regarding the difference of advantages and disadvantages of the two databases, comparing the survival of the KONOS and KOTRY database may value of comparing initial and recent period of pediatric LT in Korea.

We agree that there is bias as the data of overlapping 38 patietns are counted double. However, we believe it is still worth it because it may show if there is any era effect .

  1. Could the authors provide waiting list analysis? How many patients were listed, what was waitlist mortality, and was there any trend towards improved waitlist outcomes incorporating more deceased donor liver transplants and splitting livers?  

--> We are sorry we cannot provide waiting list analysis. The data we used includes only patients who eventually underwent LT. Thus, it is not possible to analyze mortality during waiting list. Regarding your valuable commnet, we will plan further study using data which may include waiting list analysis.

  1. the authors suggest that LDLT had worse outcomes due to higher "acuity" patients. Could the authors have performed a subgroup propensity analysis to determine that this is really true? If we controlled for equal PELD, equal acuity patients on the waiting list, would outcomes for LDLT vs. DDLT be the same?

-->KONOs database deos not include PELD score and thus, it is impossible to perform subgroup analysis. KOTRY database includes PELD score. However, according to KOTRY database, there was no significant survival difference between LDLT and DDLT (P=0.254). A longer follow-up with the KOTRY database may determine the fact clearly through further studies.

  1. the design of this study does not support their conclusions. While the observation of improved outcomes for higher volume centers is made, this is not appropriately studied or presented by the authors. A more adjusted analysis should have been done comparing outcomes for high volume centers vs. low volume centers, and their data presented. This analysis should have also been done for smaller bodyweight recipients and patients without biliary atresia. It is not enough to do the univariate analysis to arrive at this conclusion.

-->Not only univariate analysis but also further multivariate analaysis showed that high-volume center was associated with better survival (P<0.001) as described in the “Result” section (line 124-125). According to the reviewer’s comment, we again analyzed the survival for smaller bodyweight recipients (<6kg) and patietns without biliary atresia. The survival of smaller bodyweight recipients was better in high-volume center compared to that in low-volume center (P=0.003) and the survival of patients without biliary atresia was also better in high-volume center compared to that in low-volume center (P<0.001).